# Smartphone-Based Multiplexed Biosensing Tools for Health Monitoring

**DOI:** 10.3390/bios12080583

**Published:** 2022-07-29

**Authors:** Tutku Beduk, Duygu Beduk, Mohd Rahil Hasan, Emine Guler Celik, Jurgen Kosel, Jagriti Narang, Khaled Nabil Salama, Suna Timur

**Affiliations:** 1Silicon Austria Labs GmbH: Sensor Systems, 9524 Villach, Austria; tutku.beduek@silicon-austria.com; 2Central Research Test and Analysis Laboratory Application and Research Center, Ege University, 35100 Izmir, Turkey; 91200000446@ogrenci.ege.edu.tr; 3Department of Biotechnology, Jamia Hamdard, New Delhi 110062, India; rahilhasan789@gmail.com (M.R.H.); jagritinarang@jamiahamdard.ac.in (J.N.); 4Department of Bioengineering, Faculty of Engineering, Ege University, 35100 Izmir, Turkey; emine.guler.celik@ege.edu.tr; 5Sensors Lab, Advanced Membranes and Porous Materials Center, Computer, Electrical and Mathematical Science and Engineering Division, King Abdullah University of Science and Technology (KAUST), Thuwal 23955-6900, Saudi Arabia; khaled.salama@kaust.edu.sa; 6Department of Biochemistry, Faculty of Science, Ege University, 35100 Izmir, Turkey

**Keywords:** smartphone-based detection, multiplexed detection, electrochemical sensor, clinical diagnosis, biomarkers, biosensors, point-of-care (PoC) testing

## Abstract

Many emerging technologies have the potential to improve health care by providing more personalized approaches or early diagnostic methods. In this review, we cover smartphone-based multiplexed sensors as affordable and portable sensing platforms for point-of-care devices. Multiplexing has been gaining attention recently for clinical diagnosis considering certain diseases require analysis of complex biological networks instead of single-marker analysis. Smartphones offer tremendous possibilities for on-site detection analysis due to their portability, high accessibility, fast sample processing, and robust imaging capabilities. Straightforward digital analysis and convenient user interfaces support networked health care systems and individualized health monitoring. Detailed biomarker profiling provides fast and accurate analysis for disease diagnosis for limited sample volume collection. Here, multiplexed smartphone-based assays with optical and electrochemical components are covered. Possible wireless or wired communication actuators and portable and wearable sensing integration for various sensing applications are discussed. The crucial features and the weaknesses of these devices are critically evaluated.

## 1. Introduction

Laboratory health diagnostic systems have advanced over time, utilizing a range of transducing processes such as optical, magnetic, and electric field effects. Many of these technologies are difficult to utilize at the point of service because they need specialized laboratory equipment and laboratory specialists [1]. The collected patient samples frequently require pre-processing, such as isolating serums from whole blood samples, before analysis [2]. Therefore, it is necessary to develop relatively practical methods for detection of biomarkers associated with diseases in resource-limited conditions or non-clinical settings [3]. Biosensors are analytical tools that are useful for detecting specific targets by recognizing the chemical or biological process of potential disease biomarkers. A biosensor consists of a molecule (either natural, synthetic, or bioinspired) that is a bioreceptor with a high binding affinity toward a specific substance, a target/analyte, enabling its biorecognition, a transducer for converting this biorecognition event into a measurable signal, and an electronic circuit for signal processing, data evaluation, and transfer [4]. The signal generated through a biological event is transmitted in the form of electrochemical, electrochemiluminescent, magnetic, or optical signals. Biosensors play an essential role in personalized health monitoring, especially when combined with smartphones. Disease biomarkers include a variety of micromolecules, proteins, nucleic acids, whole cells, and pathogens, as well as metabolites, ions, and volatile chemical compounds. In the case of certain diseases, detecting a single marker may not be efficient enough for accurate diagnosis. The importance of multiplexed detection and point-of-care personalized health monitoring is discussed below.

### 1.1. Importance of Multiplexed Biosensing

Multiplexing is the practice of detecting or recognizing numerous biomarkers in one diagnostic test, which can be useful for a variety of illnesses. Previous studies proved that biosensors are powerful tools for the purpose of health monitoring [5]. However, focusing on single-biomarker detection is frequently insufficient to give the required information for clinical diagnosis or disease tracking [6]. In some cases, the existence or severity of certain diseases may only be identified by monitoring the concentration of multiple biosensors. Multiplexed sensors can detect a panel of discriminative biomarkers simultaneously, which can enhance detection accuracy and enable early disease diagnostics. Thus, the development of next-generation sensors has centered on multi-detection approaches [3].

The quantities of discriminative biomarkers often fluctuate at different stages for a particular illness. The diagnostic procedure for cardiovascular diseases is a common example. In particular, the response to various drugs collected from patients with cardiovascular diseases is often affected due to polymorphisms [7]. Similarly, cancer types and benign tumors may share biomarkers. Detecting several biomarkers is required for a precise cancer type identification. Particularly, breast cancer patients have considerably higher levels of many biomarkers than patients with a benign breast tumor illness [8]. Despite the fact that medical treatments have advanced significantly, there is still a great demand for advanced and practical medical services. Delayed diagnosis and treatment, especially in locations with inadequate medical resources, often result in the worsening of conditions. Moreover, performing diagnosis to detect as much information as possible from a limited sample volume is essential. In this context, multiplexed bio-(chemical) sensors are an obvious and attractive approach, specifically for limited sample volumes.

### 1.2. Point-of-Care (PoC) Personalized Health Monitoring

As an alternative to rather expensive and bulky, traditional, laboratory-based technologies, such as chromatography and spectrophotometry, biosensors based on optical or electrochemical analysis techniques have been used widely in the past decade [9,10]. The possibility of operation without professionally trained users and also the short turnaround time of these devices accelerated the use of biosensors for health monitoring. The integration of biosensors into point-of-care (PoC) testing platforms enhances the conveniency and practicality of the diagnosis. PoC assays can be based on various physical/chemical transducer types, such as enzyme, antibody, or DNA types, converting the biological signals into a response of multiple bioreceptors. The difference in the physical or chemical signal in the presence and absence of biomarkers is then converted to an electrical output on a display.

Compared to laboratory-based diagnostic methods, optical or electrochemical biosensors with a smartphone attachment are more suitable for PoC use for on-site disease diagnosis and personal health monitoring [11,12,13]. Figure 1 shows the process of diagnosis using smartphone-based biosensors. They are an effective alternative, offering tremendous potential for improving diagnosis and treatment. The present decade has seen the development of biochemical sensors integrated into smartphones and Internet of Things (IoT) platforms, which are often utilized for human health PoC testing [14]. These are often portable and cost effective, and they do not require any specific training to use [15].

## 2. Smartphone-Based Sensing Methods

In the 2000s, the data transfer capabilities of mobile phones began to be utilized in point-of-care diagnostics. Smartphones gained the ability to provide biosensing technologies through smartphone applications and tools, such as the camera, wired peripherals (USB interface and audio port), and wireless peripherals (Wi-Fi, Bluetooth, and NFC) [16]. In the last two decades, researchers started integrating analytical tools into smartphone systems for data processing and communication based on colorimetric and electrochemical detection techniques. Colorimetric detection was made possible by mobile phone cameras [17]. Other technical capabilities of smartphones in the biosensing field include data displaying and processing, depending on the detection method, to meet different points of need [5,18,19]. Due to their quality and resolution, imaging capabilities were adapted to molecular sensing [20]. Smartphone integration for biosensing makes portable and affordable real-time monitoring possible, helping to reduce the cost and complexity of sensing systems [21]. These platforms are utilized for health care applications such as surgical diagnosis and self-diagnosis [22]. Patient care has evolved into a self-monitoring platform for various physiological conditions, such as blood pressure, pulse rate, and body weight. In addition, monitoring various biomarkers, antibodies, enzymes, pathogens, and metabolites is becoming accessible for patients without the need for a health professional [23,24].

Wide data accessibility is a key factor for smart diagnostic devices. Possible integration of artificial intelligence, machine learning, and other promising technologies into the sensing platforms enhances the accessibility of smartphone-based multiplex devices [25,26]. A focus on the design and development of mobile diagnostic devices provides not only versatility, but also measurable benefits in terms of the cost of diagnosis and medical treatment. The latest smartphone-based sensor studies focused on multi-analyte sensing in various biological media to diagnose diseases in real time and monitor biomarkers simultaneously [27]. The smartphone-based detection techniques and tools with high potential for implementation in multiplexed systems are covered in this section.

*Camera and smartphone applications:* Colorimetric-detection-based smartphone sensors are commonly used as the built-in cameras play a detector role to identify the signal output. For instance, a colorimetric capillary chip was designed by Machado et al. for multiplexed immunodetection of mycotoxins. The smartphone camera was used for signal acquisition, and a simple grayscale quantification procedure was applied for the data analysis [28]. Another sensor was proposed by Khoshfetrat et al. for the simultaneous visual detection of thyroid cancer suppressor genes. Electrochemiluminescence signals were recorded on a smartphone camera with 90% sensitivity [29]. In another study, simultaneous identification of multiple bacteria was demonstrated based on a signal readout from a smartphone camera and a lateral flow assay (LFA) [30]. Overall, there is a wide range of smartphone options and attachments offering easy imaging-type diagnostics that can be implemented into a commercial product in the near future. The main actuation components of smartphones often used in biosensing applications are discussed below.

*Wired peripherals: USB interface and audio port:* Data transfer or signal transmission can be performed through a USB interface and audio port. Many studies reported that it is still a very practical way to convert a smartphone into a practical diagnostic system [31]. Sun et al. developed a module which can be directly plugged into a smartphone through the audio jack [32]. Another system was designed by Jiang et al. with a microfluidic chip and embedded circuit for communicating with the sensor through a USB port [33]. Such handheld systems provide rapid and real-time measurements, data transfer, and storage in secured cloud servers. Though these systems provide practical solutions for electrochemical systems, the presence of wires during measurements may not be completely convenient for every biosensing application.

*Wireless peripherals: Wi-Fi, Bluetooth, and NFC:* Sensor implementation has evolved around wireless technology to enhance the practicality of diagnosis. Biosensors with a wireless connection eliminate the need for cables for communication and/or power. Wi-Fi, Bluetooth, and near-field communication (NFC) are the different wireless communication protocols of current smartphones. A Bluetooth connection is employed in the biosensing field due to its high compatibility with practically all kinds of cellphone, independent of model or brand. Recently, a laser-scribed, graphene-based biosensing platform was wirelessly combined with a custom-made electrochemical analyzer (KAUSTat) through Bluetooth for the management of the pandemic. Due to its applicability to multiple amperometric and voltametric measurements and direct connection to mobile application software, it could be implemented in multiplexed platforms for continuous health monitoring [34]. Among the communication modules, NFC-based peripherals are promising for contactless biosensing technology in terms of low-power data transfer. New sensing opportunities previously considered to be conceptually and financially impossible are made possible by the coupling of NFC-based wireless power and data transfer with affordable electronics and sensors [35]. Wi-Fi is another wireless peripheral that has a bigger service area, better bandwidth, and a high prevalence in buildings and cities. Wi-Fi connection allows portable devices to communicate with any other internet-connected device. Several multiplexed diagnostic platforms were reported recently for in situ wound monitoring [36] and COVID-19 diagnosis [37], which is discussed in later sections.

### 2.1. Optical Sensors

Optical sensing is a technology that enables the real-time, label-free detection of a wide range of biological and chemical compounds. The operating principle of optical sensing is the detection of color changes in tubes, capillaries, microfluidic chips, filter paper, and nitrocellulose strips, depending on the analyte concentration. High sensitivity, biocompatibility, and easy integration into flexible platforms can be listed as some of its advantages [38]. Integrating optical sensors into smartphones has become popular in recent years due to the practical image analysis provided by smartphones [38,39,40,41]. Smartphone applications, such as FLASK station and ImageJ, are often used for data analysis [42,43]. In 2021, Zhang et al. designed a portable, optical, smartphone-based quantum barcode imaging platform that can diagnose in real time the severe acute respiratory syndrome virus (SARS-CoV-2) at different levels of infectious severity with 90% sensitivity and 100% specificity [44].

Optical sensing on smartphones is mainly classified into three categories: fluorescent, colorimetric, and chemiluminescence-based detection [24]. In order to determine the analyte concentration, a colorimeter examines the amount of light transmitted through a sample at a certain wavelength [45]. The color intensity is measured using the naked eye or a colorimeter in traditional colorimetric analyses. However, naked-eye detection has low reliability and selectivity. Commercial colorimeters are inconvenient to use daily and come with high costs. Consequently, smartphone-based colorimetric sensors are promising for rapid sensing and quantifying substances [46,47]. Color identification can be achieved by a smartphone camera and high-performance processer without the use of additional equipment (Figure 2) [48]. Recently, target analytes, such as pathogens, the SARS-CoV-2 virus, Zika virus, metal ions, organic pollutants, and hormones, were detected by colorimetric sensors [24]. For instance, Fabiani et al. developed a smartphone-based colorimetric immunoassay for SARS-CoV-2 detection in saliva which used a 96-well, wax-printed paper plate for color visualization [49]. Moreover, differentiating multiple biomarkers simultaneously with optical tools can provide fast information about diseases. Yin et al. reported a multiplexed optical sensor smartphone application for detecting SARS-CoV-2 along with other pathogens [50]. Similarly, Alawsi et al. proposed a colorimetric sensing platform Android application for on-site testing of glucose, triglycerides, and urea [51]. Fluorescence-based methods also have great potential when combined with a smartphone. The camera records the emitted light, and the light emission of a target sample is evaluated, leading to a high selectivity [52,53]. Tetracycline, a widely used antibiotic, was detected by a smart, ratiometric fluorescent sensor created by Wang et al. [54]. Chemiluminescence analysis is another optical sensing technique that uses light to determine the concentration of a chemical element or chemical compound in a solution without a light source and spectroscopic system [48]. Recently, a chemiluminescence smartphone sensor was developed by Kholafazad-Kordasht et al. for simultaneous clinical diagnosis of salivary cortisol, valproic acid, and coronavirus [24]. Another example of chemiluminescence imaging was reported by Li et al. for simultaneous cancer diagnostics using a smartphone combined with a microfluidic chip [55]. The findings confirmed that optical sensors are useful for disease identification and quantification when combined with smartphones. Without complicated instrumentation, multiplexed optical sensors combined with smart detection and screening tools save time and money as practical diagnostic tools [56,57]. However, there are several drawbacks to smartphone-based optical sensors, such as the required optimization of lighting conditions [58]. Measurements in diverse lightning conditions may be challenging considering the dependance of high-resolution cameras on specific lighting conditions. The image processing algorithm from the color space of smartphones may be affected by ambient light. Adding photo auxiliary devices placed in front of camera, such as light diffusers, prisms, etc., may eliminate the error from differences in illumination. [38]. Bergua et al. designed a smartphone-based optical sensing systems device with a portable external microplate reader that supports colorimetric, fluorescence, and luminescence techniques to detect the intensity of commercial fluorescent organic dyes when using the ImageJ application [59].

### 2.2. Electrochemical Sensors

Electrochemical reactions caused by the chemical or biological interaction between the sensing surface and the analytes are measured by electrochemical sensors. The response is converted to qualitative and quantitative electric signals which can be based on amperometry, potentiometry, and conductometry measurements (Figure 2) [40,60]. They are a well-known group of sensing methods because the techniques and equipment needed are straightforward. To date, various electrochemical systems for a wide range of analytes, including hormones, viruses, disease indicator biomarkers, and metabolites, have been reported [61,62]. With the help of smartphones, electrochemical sensors are becoming practical biomedical testing tools. The first application of a smartphone-based electrochemical sensor system for biochemical detection systems was described as amperometric sensing [63]. During amperometric sensing, constant voltage is applied to the working electrode, followed by the measurement of the current provided by the oxidation/reduction of an electroactive analyte [64]. Amperometric sensing can be chosen for use in the study of microscopic domains, detection in microflow systems, single-cell processes, and in vivo monitoring of neurochemical events. For instance, Liu et al. reported an amperometric platform combined with a portable biochip, a Bluetooth transmission system for simultaneous insulin, and glucose monitoring in saliva for practical diagnosis of diabetes and other insulin-resistance-associated diseases [65]. Another electrochemical technique, voltammetry, was also used in smartphone electrochemical sensor measurements [64,66]. The current response of a biological or chemical process was measured under an applied potential difference. Different voltametric measurement types, such as differential pulse voltammetry (DPV), cyclic voltammetry (CV), square wave voltammetry (SWV), and linear sweep voltammetry (LSV), are used by smart electrochemical sensors for various applications [67]. Along with amperometry and voltammetry, impedimetric measurements are also widely used by electrochemical sensors. Impedimetric detection, also known as electrochemical impedance spectroscopy (EIS), determines the impedance of the system as a function of frequency [66]. EIS-based electrochemical sensors were combined with smartphones in previous studies [68]. Talukder et al. reported a smartphone-based microfluidic impedance cytometer for personalized monitoring of blood cell count [69]. Another impedimetric printed electrochemical sensor was combined with a smartphone interface by Rosati et al. for rapid detection of neutrophil gelatinase-associated lipocalin in urine [70]. The overall response obtained from the electrochemical sensor was transferred via Bluetooth to a mobile application. Electrochemical sensing systems are promising for smartphone integration as they have high simplicity of fabrication [70,71,72]. Thus, it is expected that multiplexing smartphone-based electrochemical systems can accelerate accurate diagnostics, allowing the detection of various biomarkers simultaneously by using amperometric, voltametric, or impedimetric detection techniques.

In the case of voltametric sensors, the current is measured over a controlled potential variation. The voltammetry technique includes highly sensitive signal-generating or -amplifying composites for the immunoaffinity layer; therefore, it might be useful for signal identification in multiplexed systems. Impedimetric sensors provide a small amount of magnitude voltage while the frequency varies. This methodology is mainly used for characterizations of electrode surfaces and applications focusing on microorganism monitoring, pathogen detection, corrosion monitoring, and heavy metal ion detection [73]. Overall, the impedimetry method is less destructive for the electroactive area of a sensor in biological applications compared to CV and DPV [74].

The excellent detectability, high sensitivity, ease of use during experiments, and low cost of electrochemical POC devices make their integration with smartphones a very appealing method. Additionally, it is practical to develop smart electrochemical sensors that can detect biomarkers in whole blood, serum, urine, saliva, or sweat in order to prevent the unnecessary transportation of patients to hospitals. Connecting electrochemical sensors to smartphone devices enables protection from infectious diseases and remote control of chronic diseases by doctors. Despite all these advantages of smartphone-based electrochemical sensors, some challenges need to be considered. One of the constraints of electrochemical sensor–smartphone integration is the compatibility between the port type of the phone and the potentiometric device, which relies on the model and brand of the phone. Moreover, the smartphone should have certain camera features, such as reduced pixel size and improved pixel intensity, and a more powerful processor and high-speed wireless connection to be able to give accurate results.

## 3. Smartphone-Based Biosensing Technologies

Smart health diagnostic tools integrated into electronics in wearable/portable ways hold great potential in empowering patients with real-time measurements. Biomarker concentration tracking at trace levels in various body fluids such as blood, sweat, tears, and saliva is made possible using smartphone integration. A wide range of applications, from disease diagnosis and management to health monitoring, proved the widespread applicability of smart PoC testing [75]. The main smartphone-based biosensor applications are covered in this section in terms of user practicality (Figure 3); they are portable and wearable. Moreover, the importance of implementing these technologies in multiplexing platforms is emphasized.

### 3.1. Portable Biosensors

Biosensors were first designed for in vitro testing in laboratory settings. Knowing that various new technologies are generated in the biosensing field every year, there is also a need for the development of more compact and portable biosensors. The ability to use the PoC devices outside of the laboratory is essential, especially for patients living in areas that have no direct access to early and rapid disease diagnostics. Moreover, in the case of contagious diseases, the availability of portable biosensors has a high impact on the prevention of the spread of disease by reducing the travel need of infected individuals. The combination of portable biosensors with smartphones has led to various non-invasive testing applications, namely, those that use saliva, urine, and exhaled gas as target analytes [76].

*Lateral flow assay (LFA):* These paper-based platforms are useful for the detection and quantification of analytes in complex mixtures. After placing a collected sample on a LFA platform, results are displayed within 5–30 min [77]. The detection often depends on a smartphone application without any further energy sources or equipment. LFAs are developed on a strip with a nitrocellulose membrane and require biological or chemical components depending on the application. The nitrocellulose membrane consists of test lines and control lines for qualitative analysis of a disease. As the sample liquid flows through the membrane strip, pre-immobilized reagents in specific sections of the strip become active [78]. Kim et al. constructed a portable LFA using chemiluminescence for stress hormone (cortisol) detection in human serum [79]. In this work, the LFA membrane consisted of gold nanoparticles conjugated by antibodies against cortisol. Another study reported a LFA platform made of ligand-coated quantum beads and specific nanobodies for early-stage diagnosis of malaria and cardiovascular diseases [80]. Both examples included smartphone integration for data visualization and analysis. Overall, use of LFA-based devices can easily by expanded to detect simultaneous disease markers by conjugating several antibodies in one platform. This shows that the LFAs are a promising candidate for portable multiplexed applications [81]. Overall, LFAs are rapid, cheap, portable, and simple tests with a long shelf life and no need for refrigeration during storage. They can be easily adapted to daily use as point-of-care tests and can detect simultaneous disease markers by conjugating several antibodies in one platform. This shows that the LFA is a promising candidate for portable multiplexed applications. However, the fundamental disadvantage of LFAs is the lack of quantitative results. Integrating LFAs into smartphones using scanners or cameras requires specialized software to operate. Moreover, smartphone camera and flash limitations and the paper platform may affect the reliability of results while converting data to digital form.

*Lab on drone (LOD):* Drones can make health care more accessible to patients who would otherwise be unable to obtain it because of their location, a lack of resources, or both. Similarly, they can provide fast and practical monitoring of diseases. Hardy et al. developed a technique for mapping water bodies in Africa for mosquito vector-borne disease elimination [82]. Another lab-on-drone example was reported by Apprill et al. for the non-invasive exhaled breath collection of whales used to perform microbiome analysis [83]. Although, they are not widely used today, lab-on-drone applications provide an innovative approach and a fast and practical solution for nonassessable areas when interfaced with smartphones camera and digital computing power [84,85]. Priye et al. developed an in-flight PCR model for Ebola virus detection for remote area applications. This nucleic acid analysis system built on a drone enabled conventional laboratory protocols to be followed with little or no modification [86]. Thus, rapid in-flight tests with a smartphone connection can reduce the delays between sample collection and processing, allowing test results to be given in minutes. This raises the possibility of integrating biosensors into drone-based systems for simultaneous marker detection. However, drone technology still requires further development to ensure that it can prevent future incidents that threaten human lives and security of data. Another disadvantage is that the lifespan of drones is less than conventional cameras since the powering methods are limited for drones.

### 3.2. Wearable Biosensors

To date, various physical sensors have been developed as wearable or lab-on-body systems interfacing directly with the skin for the monitoring of vitals, movement, steps, calories burned, or heart rate. However, recent advancement and breakthroughs in the non-invasive monitoring of biomarkers, such as metabolites, microorganisms, and hormones, proved that biosensors can be integrated into wearable systems [87,88]. The ability to monitor various non-invasive body fluids in real time is the key advantage of wearable biosensors [89,90]. Combined with skin-mounted systems, smartphone integration is essential for real-time monitoring. However, continuous or remote monitoring, dependence on laboratory environment, and multistep procedures for sample preparation remain as challenges for smart, wearable biosensors [91]. Since biological, non-invasive fluids contain a wide range of markers, ions, macromolecules, and microorganisms that may be direct or indirect indicators of diseases, these personalized devices can be integrated into multiplexed platforms for simultaneous health monitoring [92].

*Mouthguard:* Saliva is complex media containing various micro- or macromolecules such as ions, proteins, microorganism, sugar, etc. Mouthguard sensors were investigated for real-time measurement of salivary chemicals [93]. These wearable devices are commonly used by players in both competitive and leisure sports because they provide protection against sports-related oral injuries. Researchers studied the possible applications of a mouthguard, such as the continuous wireless monitoring of pH, glucose, lactate in saliva, and salivary uric acid in oral cavity, with a smartphone operation system [94,95]. Mannoor et al. demonstrated a graphene nanosensor for continuous microorganism growth monitoring in tooth enamel [96]. As expected, achieving remote powering and wireless readout was essential for sensors implemented into mouthguards [97]. Though existing smart mouthguard sensors focus on single-biomarker detection, monitoring multiple salivary markers can lead to practical and personalized health monitoring without a need for health professionals every time. Despite its limitations, such as being prone to interference from contaminants, including charged ions, enzymes, and microorganisms, difficulties with potential toxicity and biocompatibility, and the need for real-time clinical mouth testing on humans, the smart mouthguard sensor is a useful device for simultaneous non-invasive monitoring of salivary metabolites for daily health care applications. Miniaturization and integration of circuits and power supply is the most challenging issue that needs to be solved for smartphone integration and reliable results.

*Contact lens device (LOC):* The monitoring of physiological chemicals related to health conditions and disease situations is possible by directly using the surface of the cornea where there is access to possible markers in tears, such as ions, proteins, etc. Tear fluid is commonly collected by either filter paper or glass capillary pipettes [98]. Soft bioelectronics have recently received a lot of attention for its potential use in intrinsic polymer characteristics and organic electronics for wearable and implantable health care systems. Smart contact lenses, among other wearable health care technologies, have gained significant interest. The corneal surface provides a non-invasive interface, providing connection to physiological conditions in the human body such as those in the brain, liver, heart, lung, and kidney. Recently, remote diabetes monitoring was achieved by smart wearable sensor systems integrated into soft contact lenses to evaluate the resistance change of graphene sensors upon glucose binding. In conjunction with Novartis, Google created the Google Lens for diabetes diagnostics [99]. Keum et al. constructed a non-invasive/wireless smart contact lens for diabetes diagnosis as well as the therapy of diabetic retinopathy [100]. Contact lens sensors are simple-to-use point-of-care sensing devices with a short readout time. However, there are several difficulties to overcome since the contact lenses are in constant contact with the cornea of the eye. Another important point is that the fabrication of the electrochemical sensors connected to contact lenses is quite complicated. The lens is made of a biocompatible, transparent, and conductive polymer consisting of flexible and ultrathin electrical circuits and a microcontroller allowing efficient control of drug delivery and wireless management and data communication through smartphones. Though multiplexing smartphone-based contact lens systems can provide information through simultaneous detection of ions or proteins existing in tears, the integration of electrochemical or optical sensing systems into contact lenses still requires further optimizations to achieve reliability.

*Sweat analysis patch:* Sweat is known to be a biofluid with a rich composition including ions, proteins, amino acids, metabolites, and a trace amount of cancer indicators. Due to its rich nature, researchers have tended to build multiplexed sweat monitoring devices for accurate analysis. Combined with pH and ionic strength monitoring, simultaneous measurements of certain biomarkers in sweat provide accurate information for disease diagnostics. Patches often have hydrophilic fillers for sweat collection to be transferred to the microfluidic channels for sensing measurements [101]. He et al. developed a silk-based multiplexed patch for simultaneous electrochemical detection of ascorbic acid, uric acid, glucose, and lactate, as well as potassium and sodium ions in sweat [102]. Another patch sensor was reported based on MXene/Prussian blue for simultaneous glucose and lactate detection [87]. Both examples provided real-time analysis of data wirelessly transmitted to a remote mobile device and displayed in a custom-developed smartphone application. Optical detection is another possibility for the construction of a smart sweat patch. A stretchable sensing patch for epidermal sweat analysis was developed by a superhydrophilic assay for sample collection; reference colors and colorimetric assay were decorated with silica nanoparticles (NPs) [103]. Hence, the use of in situ wireless sweat patches was successfully demonstrated for sweat analysis after physical exercise and resting sweat analysis. Various biosensing techniques are effectively in use for personified health monitoring and clinical diagnostics. Overall, a sweat patch is a practical, wearable sensor for a single use and can be considered as disposable due to the low cost of the materials. Fabrication processes may include challenges such as the complexity of sweat composition and measurement errors due to changing ambient conditions. The complexity of sweat composition and measurement errors due to the changing environmental conditions can still be challenging for sweat patch sensors. The battery of the sensor can be developed to be self-powered, supplying energy and sensing capabilities to a battery-free, reusable electronic reader, which makes the integration of a patch sensor into a smartphone easier.

## 4. Smartphone-Based Multiplexed Biosensing

### 4.1. Metabolic Biomarker Detection

Early detection of metabolites is crucial to controlling the risk and severity to the individual. The body produces metabolites as either byproducts or intermediates of metabolic processes. Monitoring metabolite biomarkers can provide useful information for chronic disease management. Real-time patient monitoring technologies proved to be effective regarding the detection of various metabolites such as glucose, lactate, and electrolytes (sodium, potassium, chloride, and hydrogen/pH) [104]. Various concentrations of metabolite biomarkers are readily available in body fluid matrices such as sweat, tears, saliva, and interstitial fluid. Shifted values of these metabolic biomarkers can indicate acute fluctuations or chronic illnesses. Sensors with multiple-metabolite detection ability offer practicality for both the self-diagnosis of patients and health care workers. These sensors are developed as portable or wearable systems to enhance practicality. On-body or skin-interfaced wearable sensors are useful for tracking metabolites in real time. As they have non-invasive or minimally invasive sample collection methods, smart biosensors have been widely developed recently [105]. A stretchable patch sensor for epidermal collection and analysis of sweat with a customized smartphone app for the color analysis is demonstrated in Figure 4A. The patch consists of colorimetric assays and reference colors [103]. A wrist-mounted, multiplexed electrochemical platform was developed by Choi et al. to analyze glucose, lactate, potassium, and sodium in sweat [106]. Gao et al. demonstrated a real-time wound-healing device for monitoring inflammatory mediators, bacterial growth, pH, and temperature at the same time as a wearable device, shown in Figure 4B [34]. The flexible sensor and PCB includes a microcontroller, a digital-to-analog converter (DAC), and analog switches to achieve multichannel operations. Portable examples of multiplexed metabolite analyzers also exist. Figure 4C shows a monolithic, paper-based device for the simultaneous colorimetric detection of three model salivary biomarkers: glucose up to 0.18 mM, lactate up to 180 μM, and uric acid up to 0.11 mM [60]. The assay is performed by placing a single drop of saliva in the central zone of the device and exploiting controlled reactions in the microfluidic paper channels, followed by the target-induced reshaping of multibranched gold nanoparticles. Though both electrochemical and colorimetric examples were reported previously, the simultaneous monitoring of several biomarkers is critical and requires full system integration to ensure the accuracy of measurements considering the complexity of biological fluids.

### 4.2. Pathogen Detection

Biomarkers can be used to objectively assess pathogenic processes since both qualitative and quantitative detection is crucial for disease identification [107]. Viruses, such as the highly pathogenic Asian avian (H5N1) influenza A, Zika, Ebola, hepatitis B and hepatitis C, coronaviruses (CoVs), SARS-CoV-1 and SARS-CoV-2, and Middle East respiratory syndrome (MERS-CoV), cause a variety of infectious diseases that affect millions of people [49,108,109,110]. However, the identification of a single biomarker is insufficient to identify a disease. Considering that multiple biomarkers are simultaneous indicators of the development and progression of pathogenic diseases, simultaneously detecting multiple biomarkers is needed for accurate diagnosis. Since detecting more than one infectious agent simultaneously encourages self-diagnosis and reduces the need for multiple molecular tests, it provides both fast and low-cost diagnosis [111].

Multiplexing platforms for viral infections have drawn a lot of attention in recent years due to the ongoing epidemic. Because of the early symptoms, such as respiratory problems, fever etc., multiplexed sensors have the ability to differentiate them from seasonal allergies, the common cold, and other viral infections [112]. Smart biosensors for multiple-pathogen detection are often designed for personalized medicine and reduce the need for health care professionals for diagnosis. Reverse transcription polymerase chain reaction (RT-PCR) is currently the gold standard method on the market for active pathogen detection [113]. Though there are multiplexed PCR systems, they require specialized equipment and attention during handling. Other commonly used pathogen detection techniques are enzyme-linked immunosorbent assays (ELISA) and LFAs, measuring the immune response of viral infections. Both techniques perform result identification by colorimetric reaction. Multiplexed smart colorimetric sensors were reported previously for pathogen detection. A portable, smartphone-based quantum barcode serological assay device for real-time SARS-CoV-2 diagnosis at different sampling dates and infectious severity is demonstrated in Figure 5A. The device is based on a databasing app to provide instantaneous results [44].

Another multiplexed device for multiple-biomarker identification for SARS-CoV-2 diagnosis is shown in Figure 5C. The platform consists of a homemade fluorescence detection analyzer and immunoassays for detecting immunoglobulin G (IgG), immunoglobulin M (IgM), and the SARS-CoV-2 antigen simultaneously. Moreover, integration of multiple virus indicators into electrochemical detection platforms has been combined with smartphone systems in recent years. RapidPlex, a portable piece of COVID-19 at-home testing equipment was introduced in recent research as a multiplexed amperometric system (Figure 5B). This device can measure SARS-CoV-2 N protein, IgG/IgM, and C-reactive protein (CRP) simultaneously by laser-engraved graphene immunosensors combined with a wireless data transfer module [37]. Another multiplexed electrochemical platform was developed by Dou et al. for SARS-CoV-2 detection. The smart sensor recognized the S gene RNA, protein, and antibody simultaneously and collects the data wirelessly [115]. As well as SARS-CoV-2 variations, HIV, Ebola, and Zika viruses were also targeted by multiplexed sensors with smartphone integration [25,116,117]. Overall, simultaneous detection of pathogenic markers is essential for rapid and accurate diagnosis for effective treatment and early detection. However, the accuracy limits the sensor fabrication and biological surface structuring due to the possibility of cross-reactivity with other pathogens of a similar nature [1].

## 5. Challenges and Outlook for Future Perspectives

Smart multiplexed sensors provide an appealing approach for the diagnosis and monitoring of progressive diseases, especially in cases of sample-volume- or resource-limited clinical settings. Table 1 summarizes existing multiplexed sensors integrated into a smartphone platform. The possibility of implementing the sensor devices into various detection methods makes smartphone-based biosensing systems very promising candidates for next-generation personalized diagnostics for tracking self-health. Multiplexed sensors perform simultaneous tests from a limited sample volume and reduce the diagnosis time. The rapid development of diagnostic kits that were approved for emergency use during the pandemic period indicates that smart sensor systems will take their place in our daily lives in the very near future. Although smartphone-based PoC systems are starting to take their place in the market, multiplexed versions of smartphone-based biosensors are yet to be commercialized. The main reason for this is that multiplexed sensors have the tendency to suffer from cross-reactivity. Immobilizing multiple recognition units in a sensing platform leads to undesired reactions within the same biological fluid, eventually affecting the output signal. Future research should be focused on improving multiplexing performance by assessing many biomarkers simultaneously and making experimental condition optimizations. Moreover, advancing these systems with machine learning methods should make them more effective and successful diagnostic tools. With the creation of more data banks supported by clinical-evidence-based algorithms, smartphone-based multiplexed sensing devices will be very important tools for early disease diagnosis, the monitoring of treatment effectiveness, continuous health tracking, and patient-specific therapies.

## 6. Conclusions

In summary, an up-to-date survey of smartphone-based multiplexed health monitoring systems was reviewed in this paper. A wide range of biosensors with a smartphone attachment was reported. Data processing and transfer of the output signal through a smartphone promote personalized diagnostic platforms. However, the smart monitoring of the qualitative and quantitative state of various biomarkers simultaneously provides practical solutions for both patients and health care professionals. Though many of the smart biosensors focus on single-analyte measurements, certain diseases are better diagnosed when multiple biomarkers are monitored. Simultaneous measurement of multiple analytes saves time and boosts practicality during diagnosis. Current studies mainly focus on pathogenic and metabolic applications of smartphone-based multiplexed sensors. These previous reports proved that multiplexed systems are possible based on both electrochemical and optical detection techniques. Considering biosensing technologies, multiplexed sensors have high potential to be implemented in portable and wearable systems. However, they require certain improvements and are not yet complete. Although they offer the convenience of the hardware and data processing, the analytical performance of multi-electrode systems still suffers from the cross-reactivity problem. Mixed concentrations of certain biomarkers in complex biological media may lead to an undesired readout signal. Overall, multiplexed smart biosensing systems have high potential in fast and practical diagnostics when the necessary optimizations are established. In the future, the biosensor field will evolve, and smart multiplexed sensors will become simpler and more accessible to meet clinical and personalized health care needs.

## Figures and Tables

**Figure 1 biosensors-12-00583-f001:**
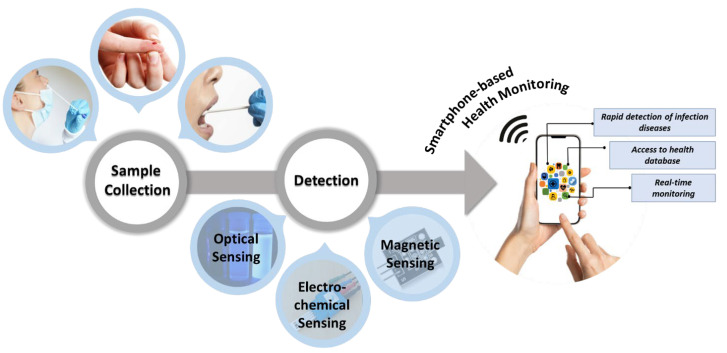
Overview of smartphone-based biosensors for health monitoring.

**Figure 2 biosensors-12-00583-f002:**
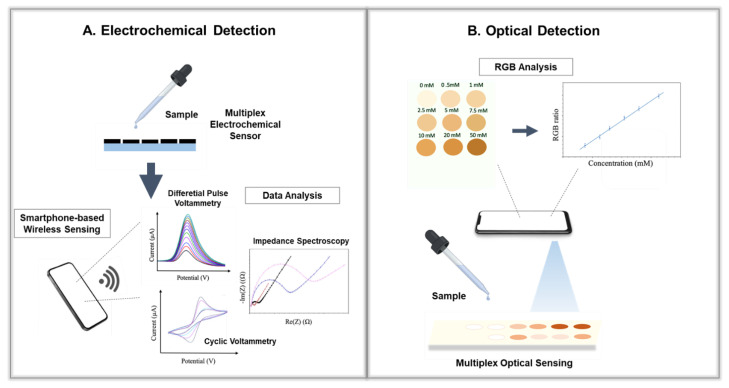
Working principle of electrochemical (**A**) and optical (**B**) detection.

**Figure 3 biosensors-12-00583-f003:**
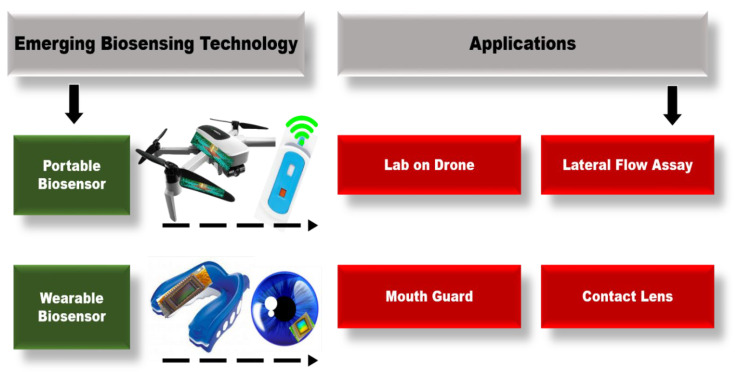
Graphical representation of the emerging biosensor technologies along with their applications.

**Figure 4 biosensors-12-00583-f004:**
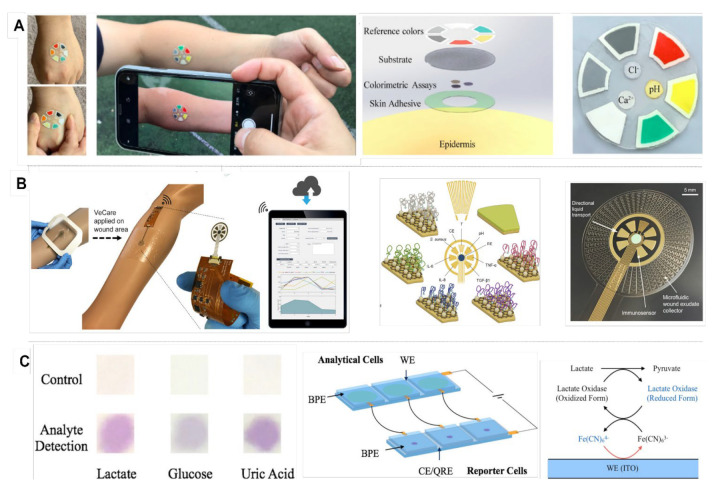
(**A**) Stretchable patch sensor for epidermal collection and analysis of sweat. Patch consists of colorimetric assays and reference colors. Adapted with permission from Ref. [103]. Copyright 2021, American Chemical Society. (**B**) A multiplex immunosensor for detection of TNF-α, IL-6, IL-8, TGF-β1, *S. aureus*, pH, and temperature for chronic wound monitoring. The device is integrated into a wireless, flexible, printed circuit board (PCB) and can be wearable. Adapted with permission from Ref. [36]. Copyright 2021, American Association for the Advancement of Science. (**C**) A closed bipolar electrode (CBE)-based two-cell electrochromic device for sensing multiple metabolites, using the simultaneous colorimetric detection of lactate, glucose, and uric acid as a model system. Adapted with permission from Ref. [60]. Copyright 2017, American Chemical Society.

**Figure 5 biosensors-12-00583-f005:**
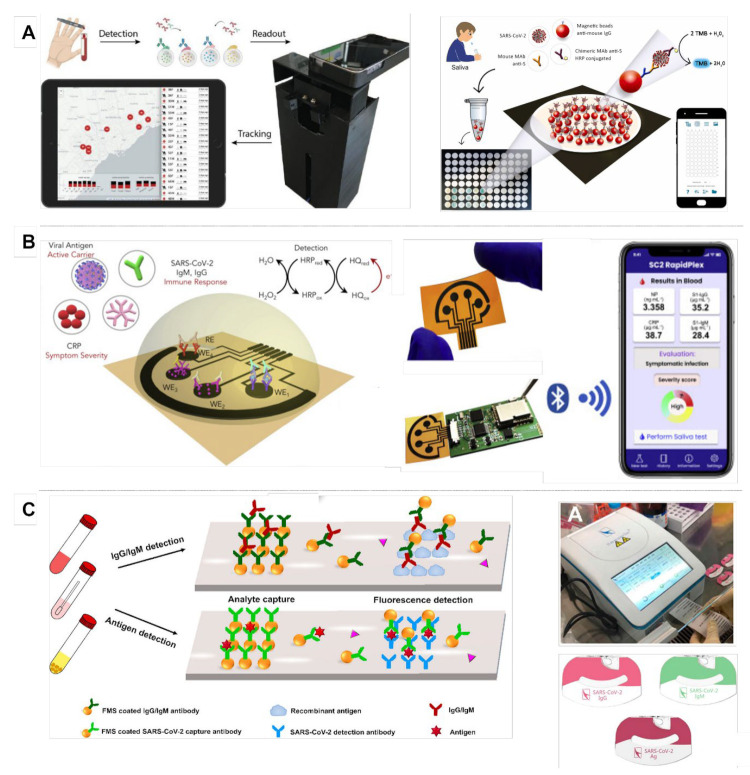
(**A**) Portable smartphone-based quantum barcode serological assay device for real-time SARS-CoV-2 diagnosis at different sampling dates and infectious severity. The device is based on a databasing app to provide instantaneous results. Adapted with permission from Ref. [44]. Copyright 2022, Elsevier. (**B**) The SARS-CoV-2 RapidPlex developed on target-specific, laser-engraved graphene immunoassays for PoC COVID-19 biomarkers. The electrochemical device is connected to a PCB and can send the signal to an app wirelessly. Adapted with permission from Ref. [37]. Copyright 2020, Elsevier. (**C**) A PoC microfluidic platform consisting of a homemade fluorescence detection analyzer, SARS-CoV-2 diagnostic microchips, and immunoassays for detecting IgG, IgM, and SARS-CoV-2 antigen. Adapted with permission from Ref. [114]. Copyright 2022, American Chemical Society.

**Table 1 biosensors-12-00583-t001:** Multiplexed PoC devices for health monitoring based on a smartphone readout.

Target Analyte	Platform	Detection Method	Application	Evaluation in Real Samples	Information	Limit of Detection	Ref
Glucose, lactate, uric acid	Paper-based carbon electrode	Closed bipolar electrode-enabled electrochromic detection	Metabolite monitoring	-	Disposable and inexpensive, high selectivity, naked-eye detection	Lactate: 180 μM Glucose: 0.18 mM Uric acid: 0.11 mM	[118]
Anti-HIV, anti-HA, anti-DEN	Microfluidic thread-based analytical device	Bioluminescence detection	Health monitoring	Human whole blood	Simple and rapid, small sample amount required, use of a 3D-printed lens adapter	Anti-HIV: 4.0 nM Anti-HA: 2.1 nM Anti-DEN: 14.9 nM	[119]
SARS-CoV-2 nucleocapsid protein, specific immunoglobulins against SARS-CoV-2 S1 spike protein and CRP	Graphene-based telemedicine platform	Electrochemical detection	Infectious disease detection	Human blood and saliva	Rapid and effective, detection of SARS-CoV-2 mutations, wireless data analysis	-	[37]
THC, alcohol	Ring-based sensor platform	Electrochemical detection	Illicit drug detection	Human saliva	Wearable wireless data analysis, rapid roadside testing, non-invasive	THC: 0.5 μM	[120]
Inflammatory mediators (TNF-α, IL-6, IL-8, TGF-β1, S)	Microfluidic immunosensing platform	Electrochemical detection	Wound monitoring	Mouse wound model	Portable wireless analyzer, flexible, non-invasive	-	[36]
Sodium, potassium, calcium, pH, uric acid, and temperature	Functionalized micropatterned-electrode array smart bandage system	Electrochemical detection	Wound monitoring	Rat wound model	High sensitivity, stability, and reproducibility, wide linear ranges, customized mobile application	-	[121]
Human coronavirus 229E, influenza A H1N1, influenza A H3N2	Air sampler with enrichment channel-integrated handheld system	qRT-PCR	Virus detection	-	Rapid and real time, requirement of additional materials for enrichment	-	[122]
Alcohol, vitamins, glucose	Wearable tear bioelectronic platform	Microfluidic electrochemical detection	Metabolite monitoring	Human tear	Wireless circuitry integrated into eyeglasses, non-invasive	-	[123]
Glucose, ethanol	Zinc oxide thin films integrated nanoporous electrode system	Impedance detection	Metabolite monitoring	Human sweat	Flexible non-invasive	Ethanol: 10 mg/dL Glucose: 0.1 mg/dL	[124]
Alprazolam, citalopram, diazepam, fluvoxamine, imipramine, nortriptyline, sertraline, zolpidem	Condition-based sensor array	Colorimetric detection	Drug monitoring	Human urine	Rapid, visual, real time, non-invasive	Flu: 0.4008 μg.mL^−1^ Nor: 0.1468 μg.mL^−1^ Cit: 0.2779 μg.mL^−1^ Alp: 0.0088 μg.mL^−1^ Dia: 0.2728 μg.mL^−1^ Ser: 0.6307 μg.mL^−1^, Zol: 0.0264 μg.mL^−1^, Imi 0.1259 μg.mL^−1^	[125]
H1N1, H7N9, H5N1	Label-free imaging array	Fluorescence detection	Health monitoring	Human serum	Good mismatch discrimination, low interference effect, early infectious disease diagnosis	H1N1: 136 pM H7N9: 141 pM H5N1: 129 pM	[126]
IL-6, thrombin	Lateral flow assays	Optical detection	Biomarker detection	-	Fast, simple, cost efficient, high sensitivity and specificity	Thrombin: 3.0 nM	[127]
HIV, leukocytosis	Giant magnetoresistive nanosensor array	Magnetic detection	Monitoring disease	Human saliva whole blood, serum	Additional circuitry, signal processing, user interface, mobile application	-	[25]
Uric-acid, nitrite, glucose	Microfluidic paper-based analytical platform	Colorimetric detection	Metabolite monitoring	-	Biocompatible ease of fabrication	Uric acid: 100 μM, Nitride: 156 μM, Glucose: 49 mg/dL	[128]
L-DOPA, tyrosine, creatinine	Periodate-modified paper platform	Colorimetric detection	Biomarker detection	Artificial urine, fetal bovine serum	Highly effective in simultaneous analysis	L-DOPA: 0.12 nM L-tyrosine: 0.93 μM Creatinine: 0.16 mg/dL	[129]
RASSF1A, SLC5A8	Fe_3_O_4_@UiO-66 and AuNRs@C_3_N_4_ NS Functionalized bipolar electrodes	Electrochemiluminescence detection	Cancer diagnostics	Cancer patient plasma sample	Monitoring therapeutic agents of patients	RASSF1A: 0.86 pM SLC5A8: 1.72 pM	[29]
Zika, Dengue, Chikungunya viruses	Complementary metal oxide semiconductor sensor	Colorimetric detection	Virus detection	Blood, urine, and saliva	Small footprint and versatility of smartphones	Zika Virus: 22 PFU/mL Dengue: 4.9 PFU/mL	[130]
Prostate-specific antigen (PSA), human chorionic gonadotropin (hCG)	Multicolor persistent luminescent nanophosphors lateral flow assay	Luminescent detection	Health monitoring	-	High sensitivity and photostability, access to minimal hardware	PSA: 0.1 ng mL^−1^ hCG: 1.0 ng mL^−1^	[131]
*Escherichia coli*, *Klebsiella pneumoniae*, *Staphylococcus aureus*	Pipette-actuated capillary array comb platform	LAMP reaction fluorescence detection	Pathogen detection	Urine	Process takes 85 min	*E. coli*: 200 copies *K. pneumoniae*: 500 copies *S. aureus*: 500 copies	[132]

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
