# Peer review of "Smartphone-Based Multiplexed Biosensing Tools for Health Monitoring"

_biosensors, 2022, doi:10.3390/bios12080583_

Round 1

Reviewer 1 Report

This review provides an overview of multiplexed biochemical sensors with integration with smart phone based signal processing and display. Overall, the paper is well-written, well-structured and provides a cohesive overview to the reader. 

These following comments should be addressed:

1. English corrections: line 110- "patient care has", line 155- "direct connection", line 331- subsection heading- "mouth guard sensors", line 344- "related to the health", line 408- "biomarkers", 

2. In lines 103-104, and 106-107, when the authors mention statements saying "detection was made possible" and "imaging capabilities were adapted", it will be good to include relevant references.

3. Table 1 is very cohesive and covers several important sensor characteristics. However, it will be good to either move Table 1 to Section 5, or refer to Table 1 in the earlier sections.

4. In Section 2.2, it will be good to conclude the section with a few sentences of discussion about which sensing mode can be useful in which scenario, and tie this discussion with the possible advantages and disadvantages/challenges of integrating electrochemical sensors with smartphone analysis.

5. Add a reference in lines 295-296.

6. It would be useful if the authors summarized the main advantages and disadvantages/challenges of each kind of assay/biosensor in section 3.1. and 3.2., and the potential difficulties of integrating smartphone technology with these sensors.

Author Response

Dear Editor,

Our responses to the comments of reviewer 1 is attached as file.

Reviewer 2 Report

The manuscript completed by Suna Timur et al reviews smartphone-based multiplexed (bio)-chemical sensing tools for health monitoring. Although subject is interesting, the contents are needed to be improved.

1.       The authors arranged the manuscript as follows: Smartphone-based Sensing Methods; Smartphone-based Biosensing Technologies; Smartphone-based Multiplexed Biochemical Sensing. The author only used a small amount of space to write smartphone-based multiplexed biochemical sensing, although both sensing methods and biosensing technologies as the basis of multiplexed biochemical sensing. Obviously, the arrangement is not reasonable.

2.       Multiplexed biochemical analysis should be stated as a separate part.

3.       What are the differences between sensing methods and biosensing technologies? The contents including camera and smartphone applications, wired peripherals, USB interface and audio port, as well as wireless peripherals (Wifi, Bluetooth and NFC), should be attributed to technologies, not methods.

4.       Where is sixth part? The authors should carefully check the whole manuscript.

Author Response

Our responses to the comments of Reviewer 2 is attached as the file.

Kind regards

suna

Round 2

Reviewer 2 Report

The manuscript can be accepted in its current state.